# Promotion Effects of Ultrafine Bubbles/Nanobubbles on Seed Germination

**DOI:** 10.3390/nano13101677

**Published:** 2023-05-19

**Authors:** Seiichi Oshita, Surina Boerzhijin, Hiromi Kameya, Masatoshi Yoshimura, Itaru Sotome

**Affiliations:** 1Graduate School of Agricultural and Life Sciences, The University of Tokyo, Tokyo 113-8657, Japan; ayoshimura@mail.ecc.u-tokyo.ac.jp (M.Y.); itarus@g.ecc.u-tokyo.ac.jp (I.S.); 2National Research Institute of Brewing, 3-7-1 Kagamiyama, Higashihiroshima 739-0046, Japan; surina@nrib.go.jp; 3Institute of Food Research, National Agriculture and Food Research Organization (NARO), Kan-nondai, Tsukuba 305-8642, Japan; hkameya@affrc.go.jp

**Keywords:** ultrafine bubbles (UFBs), nanobubbles, seed germination promotion, ROS, ESR

## Abstract

The number concentrations of air UFBs were controlled, approximately, by adjusting the generation time. UFB waters, ranging from 1.4 × 10^8^ mL^−1^ to 1.0 × 10^9^ mL^−1^, were prepared. Barley seeds were submerged in beakers filled with distilled water and UFB water in a ratio of 10 mL of water per seed. The experimental observations of seed germination clarified the role of UFB number concentrations; that is, a higher number concentration induced earlier seed germination. In addition, excessively high UFB number concentrations caused suppression of seed germination. A possible reason for the positive or negative effects of UFBs on seed germination could be ROS generation (hydroxyl radicals and ∙OH, OH radicals) in UFB water. This was supported by the detection of ESR spectra of the CYPMPO-OH adduct in O_2_ UFB water. However, the question still remains: how can OH radicals be generated in O_2_ UFB water?

## 1. Introduction

Ultrafine bubbles (UFBs), sometimes called nanobubbles, are defined as gas bubbles with a volume-equivalent diameter of less than 1 μm in ISO 20480-1:2017 [1]. The existence of UFBs and their peculiar effects has attracted increasing attention in pure and applied science and technology, covering many areas, such as the physicochemical, medical, and biological fields, among others [2,3,4,5,6,7]. In the agricultural field, accumulating evidence has suggested that UFBs and microbubbles enhance growth processes. Park and Kurata reported that microbubbles in a hydroponic nutrient solution resulted in about twice as much lettuce growth compared to that achieved with the control solution [8]. It was also reported that microbubble generation in nutrient solutions promoted lettuce growth [9]. Additionally, it was found that the fresh and dry weights of shoots were higher in tap water with UFBs than in tap water without UFBs in another application of UFBs to lettuce production in hydroponic systems [10]. A similar effect was shown by Ebina et al., where the growth of *Brassica campestris*, cultured hydroponically for 4 weeks within air nanobubble water, was significantly promoted, compared to normal water [11]. The effects of UFBs on rice production were also examined, not only in a laboratory experiment, but also in a field experiment by Wang et al. [12]. They found that UFBs stimulated gibberellin growth hormone synthesis and upregulated the plant nutrient absorption genes in rice seedlings. They also found that UFB treatment significantly increased rice yield by almost 8% more than the control, resulting in the use of approximately 25% less fertilizer, compared to the control, in a field experiment.

As seed germination is very important at the beginning stages of plant growth, the effects of UFBs on seed germination have also been investigated. We reported that UFB water exhibited a longer NMR *T*_2_ value than control water, which resulted in an increase in the mobility of water molecules and a higher germination ratio in barley seeds [13]. The promotion of seed germination due to UFBs has also been examined in several aspects, namely ROS (reactive oxygen species) production in UFB water and barley seeds [14]; identification of ROS and their effects on vegetable seeds [15]; a change in gene expression within the barley seeds [16]; the number concentration of UFBs and their effect on barley seeds [17]. Other researchers have also reported on seed germination from the perspective of different kinds of gases comprising UFBs [18] and via a comparison between various priming treatments [19].

These studies proved that the growth of plants, starting from seed germination, is promoted by UFB. This fact raises the question: what is the role of UFB number concentration in growth promotion? The answer has yet to be elucidated. As for the UFB promotional effect mechanism, it has been suggested that OH radicals—a type of reactive oxygen species (ROS)—detected using a sensitive fluorescence probe, APF, in UFB water, may be one of the factors stimulating the promotion of growth [15,18]. On the other hand, the opposite finding was reported using numerical simulation, suggesting that no OH radicals were produced from dissolving UFBs, and that hydrogen peroxide (H_2_O_2_) was produced inside an ozone or oxygen microbubble in water during hydrodynamic or acoustic cavitation due to violent collapse [20,21,22,23].

In light of this situation, in this study, the promotional effects of UFBs on seed germination were examined in order to elucidate the role of the UFB number concentration. Air and O_2_ UFB water was generated by supplying distilled water to the venturi-type UFB-generating system. As a target sample, barley seeds were selected, because barley is very well known as a model crop in plant breeding methodology, genetics, cytogenetics, pathology, virology, and biotechnology studies [24]. In parallel to germination examination, detection of OH radicals in oxygen (O_2_)-based UFB water was reattempted using ESR in order to confirm OH radical generation in UFB water with the presence of no external stimuli.

## 2. Materials and Methods

### 2.1. Seed Germination

#### 2.1.1. Air UFBs for the Use of Seed Germination and Its Measuring Device

Air UFBs, in water, were generated using an ultrafine bubble generator (GALF FZ1N-10, IDEC Corporation, Osaka, Japan), a kind of venturi-type generator [7] with a pressure dissolution system, the pressure of which (just after the pressurizing pump) was around 700 kPa, and a saturator around 300 kPa. The UFB generator was equipped with a 15 L water tank and the circulation flow rate of distilled water (Autostill WA-53, Yamato Scientific Co., Ltd., Tokyo, Japan) was 0.83 L/min^−1^. An air filter (KIC-T6, AS ONE Corporation, Osaka, Japan) with 0.01 μm filtration accuracy was mounted on the front of the gas inlet. The UFB generator was operated for 10 to 120 min to generate the approximate desired number concentrations of UFBs. After the generation of air UFBs, UFB waters were stored at 25 °C overnight in order to stabilize UFBs. They were then used for a seed germination test.

The number concentration and bubble size distribution of the UFBs were measured using a commercial device via the particle tracking analysis method (NanoSight-LM10, Quantum Design Inc., Tokyo, Japan). The measuring range of the device was between 50 and 1000 nm, with a laser light source wavelength of 635 nm and 40 mW of power, a black and white CCD mounted camera, and NTA 3.1 Build 3.1.46 analysis software. Measurements were made at room temperature, around 22 °C.

#### 2.1.2. Seed Material

To examine the fundamental effects attributable to UFB number concentration, barley seeds (*Hordeum vulagre* L., cv. Kobinkatagi) were used as the material. A germination ratio of about 100% can normally be expected due to the good quality of this type of barley seed. For this reason, the fundamental effects of UFBs on germination could be examined by simply evaluating the germination process using only T_50_, as outlined in Section 2.1.4.

Barley seeds (*Hordeum vulagre* L., cv. Yumesakiboshi) were also used to examine the effects of excess UFB number concentration. This barley seed is known to have a low germination ratio by nature, even though seed quality is good, and UFBs’ effects on germination were expected to be observed more easily according to two parameters, T_50_ and G_max_, as outlined in Section 2.1.4. There was only one report indicating the possibility of carrot seed germination suppression at higher UFB number concentrations [15]. The experimental setup in this study aimed to indicate that there was a certain upper limit of UFB number concentration beyond which UFBs negatively affected seed germination.

#### 2.1.3. Germination Test

Germination tests were performed with three seed groups for UFB water and control water sections. Each group was composed of 50 seeds in a net-like plastic bag. Thus, the total number of seeds used for UFB and control water sections was 150 (each). Each group of barley seeds was submerged in glass beakers with a volume of 2 L of distilled water (control) and UFB waters containing different UFB number concentrations in a ratio of 10 mL of water per seed. UFB waters containing 4 different number concentrations were used to examine the fundamental effects, and those containing 3 different number concentrations were used to examine the excess number concentration effect. During the germination tests, control water and UFB waters were changed twice daily to avoid a lack of oxygen and to maintain a certain amount of UFBs in water. Germination tests were performed in the dark at 25 °C. The germination ratio obtained from three independent replicates of each group was shown as the mean value.

#### 2.1.4. Analysis of Germination Process

The germination process was analyzed using a dose–response model, as follows [25]:G(t_i_) = G_max_/[1 + exp(B(log(t_i_) − log(T_50_)))](1)
where G_max_ is the maximum germination ratio, t_i_ is the time for each inspection, T_50_ is the time at which the inferred germination ratio is 50% of G_max_, G(t_i_) is the observed germination ratio for each inspection, and B is the slope at the time T_50_. We set t_0.1_ as the starting time of calculation instead of t_0_, in order to avoid the calculation of log 0 for smooth data analysis—which was conducted using VBA 7.1 developed with Microsoft Excel 2013.

### 2.2. Evaluation of ROS in UFB Water

#### 2.2.1. Oxygen UFBs for ROS Detection

The same UFB generator described above was used to generate O_2_ UFBs with the aid of the IDEC Corporation. Distilled water (FUJIFILM Wako chemicals, Osaka, Japan) was poured into the water tank and pure O_2_ was supplied from a gas inlet during the operation of the UFB generator, for about 1 h, to generate O_2_ UFBs. After O_2_ UFB water was poured into glass bottles and sealed without headspace, it was stored for 20 d at 4 °C. Then, it was concentrated using vaporization under reduced pressure to reach about 1/200 of the initial volume, according to the procedure described in the patent [26]. Foreign matter was then filtered out from the concentrated O_2_ UFB water using a polycarbonate membrane with a filtration accuracy of 0.20 μm (K020A025A, Advantec Toyo Kaisha, Ltd., Tokyo, Japan). After filtration, a high concentration of O_2_ UFB water (O_2_ UFB water has been used herein) was poured into 4 vials with volumes of 3 mL and sealed without headspace. They were then stored for another 10 d at 4 °C before ESR measurements were conducted.

#### 2.2.2. Electron Spin Resonance (ESR) Measurement

An X-band (9.8 GHz) ESR spectrometer (EMX-plus, Bruker Japan, Yokohama, Japan) equipped with 100 kHz field modulation was used to acquire ESR measurements for the detection of ROS. The spectrometer settings were as follows: resonance field of ~3372–3672 G; 1 G field modulation width; 6 mW microwave power; 0.1 s time constant. ESR spectra were accumulated at room temperature. As a spin trapping reagent, 80 mM of 5-(2,2-dimethyl-1, 3-propoxy cyclophoryl)-55-methil-1-pyrroline N-oxide (CYPMPO, MW = 247.23) was used to detect OH radicals.

O_2_ UFB water with 80 mM of CYPMPO was added to a disposable flat cell with an RDC-60-S syringe (FlashPoint Co., Ltd., Tokyo, Japan). ESR measurements were then conducted to examine the formation of adducts between ROS and CYPMPO, under the condition that no dynamic stimuli were applied to the O_2_ UFB water. Another ESR measurement was also conducted to confirm the generation of OH radicals, detecting the hydroxyl adduct of CYPMPO (CYPMPO-OH adduct) induced by the application of ultrasonic sound (43 kHz, 80 W) for 30 s to the O_2_ UFB water with 80 mM of CYPMPO in a disposable flat cell. All measurements were carried out at room temperature.

## 3. Results and Discussion

### 3.1. Fundamental Aspect of the Effect of UFBs on Seed Germination

Figure 1 shows the observed germination ratios of seeds submerged in UFB1 (□), UFB2 (◇), UFB3 (△), and UFB4 (+) water at each inspection time, together with those of seeds in control water (○). Each regression curve, indicating the seed germination processes in UFB1, UFB2, UFB3, UFB4, and control water, was obtained using Equation (1). These were denoted as A1, A2, A3, A4, and control, respectively. The number concentrations of UFB1 to UFB4 showed the relationship (UFB1 > UFB2 > UFB3 > UFB4), as shown in Table 1, together with each mean diameter.

The T_50_ of seeds in UFB1 to UFB4 are indicated in Figure 1, and the actual values of T_50_ are provided in Table 1. Their relationship was observed as (T_50_-UFB1 < T_50_-UFB2 < T_50_-UFB3 < T_50_-UFB4). A significant difference between T_50_-UFB1 and T_50_-UFB2 was observed, as supported by Figure 2. In the same way, a significant difference was also observed between and among T_50_-UFB2, T_50_-UFB3, and T_50_-UFB4. Comparing two seed groups, the group with shorter T_50_ was denoted as the group of earlier seed germination because it reached 50% of G_max_ earlier than the other group. As the T_50_-control was 16.5 h, as seen in Figure 1, with a shorter T_50_ indicating earlier seed germination, all UFB waters from UFB1 to UFB4 induced the promotion of seed germination, compared with the seeds in the control water.

Dissolved oxygen concentrations (DO) of UFB waters, measured just after they were added to glass beakers, were 9.4 mgL^−1^, 9.2 mgL^−1^, 8.8 mgL^−1^, 8.8 mgL^−1^, and 7.1 mgL^−1^ for UFB1, UFB2, UFB3, UFB4, and control water, respectively. It can therefore be said that germination promotion did not occur solely due to the high DO of UFB waters. This was supported by our preliminary experiment using another group of barley seeds, where the germination ratio of seeds in UFB water for which DO was adjusted to 8.2 mgL^−1^ was 61%, and that in control water with the same DO was 43%, at an inspection time of 12 h after submerging. In other words, UFBs affected seed germination promotion even under the same DO.

Although UFB growth promotion has been acknowledged, as described in the Introduction, the number concentration role remains unaccounted for. Regarding this issue, the results shown here presented the fundamental aspects of the effects of UFBs on seed germination. As shown, a higher UFB number concentration led to a greater seed germination promotion effect within a proper number concentration range, such as the range set in this experiment.

### 3.2. Negative Effect on Seed Germination Caused by Excess Number Concentration

Figure 3 shows the observed germination ratios of seeds submerged in UFB1 (△), UFB2 (□), and UFB3 (○) water at each inspection time, together with those of seeds in control water (●). Each regression curve indicating the germination process of seeds in UFB1, UFB2, UFB3, and control water was obtained using Equation (1); these were denoted as UFB1, UFB2, UFB3, and control, respectively. The number concentrations of UFB1 to UF3 showed the relationship (UFB1 > UFB2 > UFB3), as shown in Table 2, together with each mean diameter.

In the context of the knowledge clarified in the Section 3.1, a higher UFB number concentration was expected to lead to greater seed germination promotion. However, the seed regression curve in UFB1 water, containing the highest number concentration, appeared under that of seeds in UFB2 water, in which the number concentration was lower than that in UFB1. This was also supported quantitatively by the T_50_ and G_max_ parameters; that is, (T_50_-UFB1 > T_50_-UFB2) and (G_max_-UFB1 < G_max_-UFB2). In other words, UFB1 water had a negative effect on seed germination.

On the other hand, seed germination in UFB2 and UFB3 water followed the fundamental aspects of the effects of UFBs, as observed in the Section 3.1. The higher number concentration led to a higher germination ratio; that is, the relationship (UFB2 > UFB3) achieved results as follows: (T_50_-UFB2 < T_50_-UFB3) and (G_max_-UFB2 > G_max_-UFB3).

The results shown here meant that there was an upper limit of UFB number concentration beyond which seed germination was suppressed, and UFB1 seemed to have exceeded this upper limit. This understanding was supported by previous research in plant physiology. Bailly et al. suggested a model named the oxidative window to account for the dual role, i.e., toxic or signaling effects, of ROS generated within seeds, indicating that seed germination was only possible when the ROS content of the seed was within the range of the oxidative window [27]. Our previous paper indicated that the submerging of seeds in UFB water contributed to producing higher levels of endogenous ROS (superoxide radicals, O_2_^●−^) than doing so in distilled (control) water [14]. Considering these factors, only the number concentration of UFB1 water induced ROS in seeds in which the content was beyond the upper limit of the oxidative window, and the number concentrations of other UFB waters stimulated production of ROS in seeds, the levels of which were maintained at an amount that triggered regular cellular events associated with germination, such as hormone signaling. In other words, UFB was proven to stimulate seeds to produce endogenous ROS, thus upregulating seed germination. This had a positive correlation with the UFB number concentration, as long as the endogenous ROS were within the oxidative window.

### 3.3. A Possible Factor Promoting Seed Germination

In the Section 3.1 and 3.2, two aspects of UFB were shown. The first of these demonstrated that a higher UFB number concentration led to a greater seed germination promotional effect within a proper number concentration range, corresponding with the oxidative window. The second was that there was a UFB number concentration upper limit, beyond which seed germination was suppressed. In light of this, we posed the following question: what is the factor attributed to UFB water that affected the promotion or suppression of seed germination? A powerful clue was provided in many studies, explaining that exogenous H_2_O_2_, an ROS, played a role in promoting seed germination [28,29,30,31]. We paid particular attention to this and found, in our previous paper, that OH radicals (but not H_2_O_2_) were generated in UFB water, which stimulated the generation of endogenous ROS to promote/suppress vegetable seed germination [15]. However, negative opinions about the generation of OH radicals in UFB water, free from the influence of cavitation, have also been published [20,21,22,23]. Thus, we conducted systematic germination tests on barley seeds and found that UFB number concentration had either a positive or negative correlation with germination, depending on concentration. This suggested that the OH radicals were produced in UFB water, as described in the Section 3.4.

With the above description, we came again to the same hypothesis as in the previous paper; a very small number of UFBs disappeared continuously, and a very small amount of OH radicals were generated, which affected seed germination [14].

### 3.4. Detection of ROS in O_2_ UFB Water

The O_2_ UFB water number concentrations in the four vials were 1.1 × 10^11^ mL^−1^, 1.0 × 10^11^ mL^−1^, 1.3 × 10^11^ mL^−1^, and 1.2 × 10^11^ mL^−1^. Figure 4 shows the representative spectra observed from O_2_ UFB water, first without exposure to any dynamic stimuli (black line) and then after ultrasonic irradiation (red line). The signal intensity of the latter was reduced three times from its original signal intensity, producing a visible improvement. Both clearly demonstrated the typical spectra of the CYPMPO-OH adduct [32,33]. Similar spectra were also observed in other O_2_ UFB waters, with and without ultrasonic irradiation, taken from three different vials.

The detection of the CYPMPO-OH adduct in O_2_ UFB water after ultrasonic irradiation was a natural consequence, as the violent collapse of UFBs caused by ultrasonic irradiation resulted in the formation of OH radicals [22,23,34,35]. Interestingly, we observed the CYPMPO-OH adduct in the O_2_ UFB water without the presence of any dynamic stimuli.

In our previous paper, the OH radicals were observed in O_2_ UFB water using an APF fluorescence probe [14,15]. However, numerical simulations indicated that the ROS signals observed after the generation of UFBs did not originate in OH radicals, but instead originated in H_2_O_2_ [21,22,23]. This opinion was based on the phenomenon that an appreciable amount of H_2_O_2_ was produced from violent collapses of cavitation bubbles during UFB generation. The lifetime of OH radicals can be as short as 1 ns [36] or 20 ns [22], and that of H_2_O_2_ is generally between hours and days in subsurface [37]. Therefore, the O_2_ UFB water used in this study was stored for a total of 30 d before ESR measurements were conducted to assure the disappearance of not only OH radicals, but also the H_2_O_2_ produced during UFB generation. With this storage treatment, both OH radicals and H_2_O_2_ produced during cavitation [23] could be excluded from O_2_ UFB water used for ESR measurement. The possibility of OH radical production triggered by a chemical reaction between H_2_O_2_ and O_3_ [23] could also be excluded, because the O_2_ UFB water used at the time of ESR measurement contained neither H_2_O_2_ nor O_3_. Furthermore, it was also suggested that the OH radicals detected in the experiments could not have originated from dissolving bubbles [23]. The numerical simulations quoted here seemed convincing; however, we assumed that the detected signals observed in this study were from OH radicals that existed in O_2_ UFB water stored for a long period without being exposed to any dynamic stimuli. For this reason, the following question still remained: how can OH radicals be generated in O_2_ UFB water without being exposed to dynamic stimuli?

From another perspective, the observation of OH radicals in O_2_ UFB water without any dynamic stimuli during a long storage period provided evidence of the long-term existence of UFBs as distinct from the foreign matter inevitably found in water—although this is a qualitative estimation.

## 4. Conclusions

In this study, barley seed germination tests were systematically conducted to examine the role of UFB number concentration. It was found that UFB number concentration had either a positive or negative correlation with germination, depending on concentration. This implied that the production of ROS in UFB water could play an exogenous role in ROS stimulation, generating endogenous ROS in seeds to promote/suppress seed germination. Thus, we conducted ESR measurements and detected OH radicals in O_2_ UFB water without exposure to dynamic stimuli. Based on these results, it was suggested that UFB number concentration correlated with OH radical production, which induced the promotion or suppression of germination, depending on its content. Our findings on the role of number concentration of UFBs will provide scientific information in cases in which UFB water is applied to seeds in order to promote germination—one of many applications of UFBs in agricultural production. Finally, it will also be necessary to consider the possibility of a phenomenon in which OH radicals could be produced in UFB water separately from the process of UFB dissolution.

## Figures and Tables

**Figure 1 nanomaterials-13-01677-f001:**
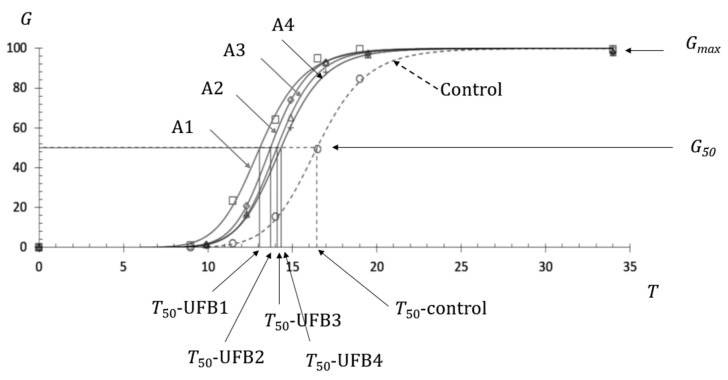
Promotion of UFBs on barley seed germination of cv. Kobinkatagi in four different UFB number concentrations in water, indicating the highest number concentration (UFB1), which results in the highest promotional effect. *G* indicates the germination ratio in %, *T* indicates time in h and A1, A2, A3, and A4 indicate the germination process of seeds submerged in water containing number concentrations of UFB1, UFB2, UFB3, and UFB4, respectively.

**Figure 2 nanomaterials-13-01677-f002:**
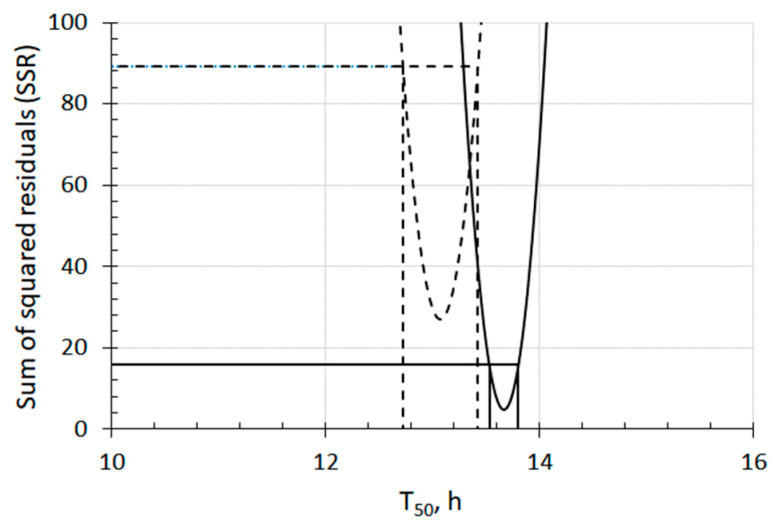
Sum of squared residuals (SSR) and 95% confidence intervals of T_50_ of both UFB1 and UFB2 barley seed sections. Dashed and solid curves show an SSR of T_50_ for seeds in UFB1 and UFB2 water, respectively. The segments of horizontal lines bounded by curves of SSR indicate a 95% confidence interval. A significant difference is observed as they do not overlap with one another.

**Figure 3 nanomaterials-13-01677-f003:**
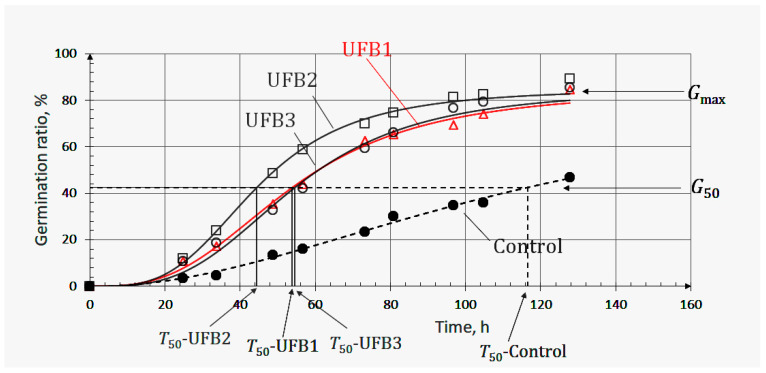
Positive and negative effects of UFBs on germination of barley seeds (cv. Yumesakiboshi) in three different UFB number concentrations in water, with the highest number concentration (UFB1, colored in red) exerting a negative effect on seed germination. UFB1, UFB2, and UFB3 indicate the germination process of seeds submerged in water containing the number concentrations of UFB1, UFB2, and UFB3, respectively.

**Figure 4 nanomaterials-13-01677-f004:**
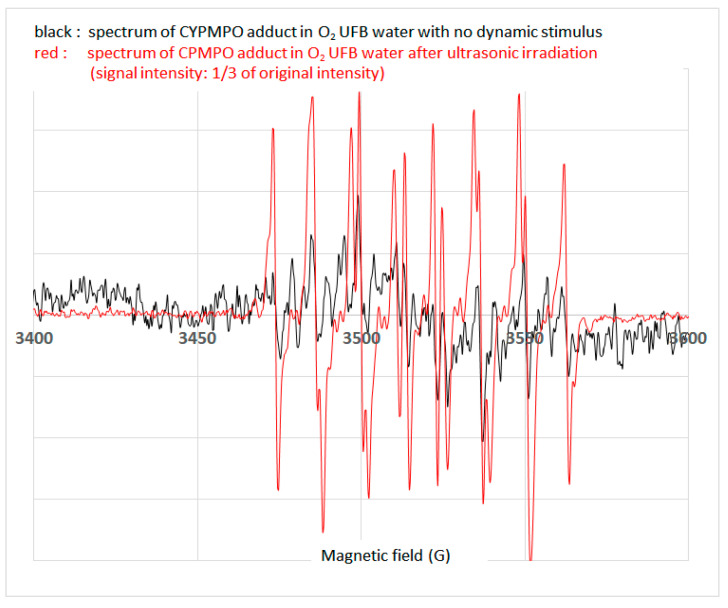
ESR spectra of CYPMPO-OH adducts observed from O_2_ UFB waters. The black line shows the signal intensity of CYPMPO-OH adduct, observed from O_2_ UFB water without any dynamic stimuli, and the red line shows the signal intensity observed from O_2_ UFB water after ultrasonic irradiation at 1/3 of its original signal intensity.

**Table 1 nanomaterials-13-01677-t001:** Characteristics of UFB water, from UFB1 to UFB4, and T_50_ of seeds germinated in each.

	UFB1	UFB2	UFB3	UFB4
Number concentration ± SD, mL^−1^	8.5 × 10^8^ ± 3.6 × 10^7^	5.0 × 10^8^ ± 3.7 × 10^7^	3.8 × 10^8^ ± 4.8 × 10^7^	1.4 × 10^8^ ± 1.7 × 10^7^
Mean diameter ± SD, nm	127.3 ± 4.7	123.5 ± 2.8	135.9 ± 4.1	132.5 ± 7.5
T_50_, h	13.1	13.7	14.1	14.3

**Table 2 nanomaterials-13-01677-t002:** Characteristics of UFB water, from UFB1 to UFB3, and T_50_ of seeds germinated in each sample.

	UFB1	UFB2	UFB3
Number concentration ± SD, mL^−1^	1.0 × 10^9^ ± 4.4 × 10^7^	7.4 × 10^8^ ± 9.0 × 10^7^	2.4 × 10^8^ ± 3.3 × 10^7^
Mean diameter ± SD, nm	136.8 ± 3.4	147.1 ± 5.4	143.6 ± 5.7
T_50_, h	53.9	44.4	54.6

## Data Availability

The data that support the findings of this study are available from the corresponding author upon reasonable request.

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
