# Peer review of "Promotion Effects of Ultrafine Bubbles/Nanobubbles on Seed Germination"

_nanomaterials, 2023, doi:10.3390/nano13101677_

Round 1
Reviewer 1 Report
1, What is the mechanism of promoting seed germination by ultrafine bubbles/nanobubbles? Free radicals? Or oxygen? The depth of research in this article is insufficient.
2, The study of seed germination experiments requires conducting experiments on multiple different plant seeds. This article only selected barley seeds, which is not scientific enough and the workload is a bit small.
3, The author is studying ultrafine or nanobubbles, shouldn't large bubbles be used as a control?
Reviewer 2 Report
The article is of the high applicable interest for agricultural production companies and farmers.
Few points must be imporved:
UFB-generating system in abstract must be described as in introduction: Ultrafine bubbles.
Line 17 - earlier seed germination - must be provided the meaning and number of germination.
Line 21-22 - However, the question still remains: how can OH radicals be generated in O2 UFB water? - Please provide the possible expantion from your side, not a question.
Generally, it must be provided a total tecxhnology which allows the agricultural production companies to use UFB treatment of plants, because there is not enough information on how this method is usually using in the industrial scale.
English language is understandable and need onle the minor corrections befor publishsing.
Author Response
Please see the attachement.

Round 2
Reviewer 1 Report
Authors have addressed all my comments.